# Meta-Inflammation and De Novo Lipogenesis Markers Are Involved in Metabolic Associated Fatty Liver Disease Progression in BTBR ob/ob Mice

**DOI:** 10.3390/ijms23073965

**Published:** 2022-04-02

**Authors:** Lucas Opazo-Ríos, Manuel Soto-Catalán, Iolanda Lázaro, Aleix Sala-Vila, Luna Jiménez-Castilla, Macarena Orejudo, Juan Antonio Moreno, Jesús Egido, Sebastián Mas-Fontao

**Affiliations:** 1Renal, Vascular and Diabetes Research Laboratory, IIS-Fundación Jiménez Díaz, Universidad Autónoma de Madrid, Spanish Biomedical Research Centre in Diabetes and Associated Metabolic Disorders (CIBERDEM), 28040 Madrid, Spain; manuel.soto.catalan@gmail.com (M.S.-C.); luna.jimenez@quironsalud.es (L.J.-C.); macarena.orejudo@quironsalud.es (M.O.); jegido@quironsalud.es (J.E.); 2Facultad de Ciencias de la Salud, Universidad de Las Américas, Concepción-Talcahuano 4301099, Chile; 3Hospital del Mar Medical Research Institute (IMIM), 08003 Barcelona, Spain; iolan.lazaro@gmail.com (I.L.); asala3@imim.es (A.S.-V.); 4Department of Cell Biology, Physiology and Immunology, University of Cordoba, 14004 Cordoba, Spain; jamoreno@fjd.es; 5Maimonides Biomedical Research Institute of Cordoba (IMIBIC), UGC Nephrology, Hospital Universitario Reina Sofía, 14004 Cordoba, Spain

**Keywords:** metabolic associated fatty liver disease, BTBR ob/ob, de novo lipogenesis, meta-inflammation

## Abstract

Metabolic associated fatty liver disease (MAFLD) is a hepatic manifestation of metabolic syndrome and usually associated with obesity and diabetes. Our aim is to characterize the pathophysiological mechanism involved in MAFLD development in Black Tan and brachyuric (BTBR) insulin-resistant mice in combination with leptin deficiency (ob/ob). We studied liver morphology and biochemistry on our diabetic and obese mice model (BTBR ob/ob) as well as a diabetic non-obese control (BTBR + streptozotocin) and non-diabetic control mice (BTBR wild type) from 4–22 weeks. Lipid composition was assessed, and lipid related pathways were studied at transcriptional and protein level. Microvesicular steatosis was evident in BTBR ob/ob from week 6, progressing to macrovesicular in the following weeks. At 12th week, inflammatory clusters, activation of STAT3 and Nrf2 signaling pathways, and hepatocellular ballooning. At 22 weeks, the histopathological features previously observed were maintained and no signs of fibrosis were detected. Lipidomic analysis showed profiles associated with de novo lipogenesis (DNL). BTBR ob/ob mice develop MAFLD profile that resemble pathological features observed in humans, with overactivation of inflammatory response, oxidative stress and DNL signaling pathways. Therefore, BTBR ob/ob mouse is an excellent model for the study of the steatosis to steatohepatitis transition.

## 1. Introduction

Non-Alcoholic Fatty Liver Disease (NAFLD), is defined as an increased ectopic fat deposition (more than 5% of liver weight), independently of excessive alcohol consumption, hepatitis C infection, antiretroviral therapy, or chemotherapy drugs. It can range from simple steatosis to steatohepatitis, characterized by liver inflammation and hepatocyte ballooning, with or without fibrosis and described as liver cirrhosis [1,2]. This condition is usually associated with obesity, systemic insulin-resistance and diabetes, being the most common cause of chronic liver disease, end-stage liver disease and liver transplantation worldwide [3,4].

The coexistence and synergistic links between the development of NAFLD and type 2 diabetes mellitus (T2DM) are well known [5,6]. Thus 56% of patients with T2DM present NAFLD [5]. But the relationship seems to be bidirectional [7,8] since in many cases NAFLD precedes and/or promotes the development of T2DM, meta-analyses have shown that the risk of developing T2DM in subjects with NAFLD doubles (HR 2.19) [9], also T2DM developments risk increasing with NAFLD severity. But also, as seen in the Korean population, if to the presence of NAFLD is added two other factors that coexist in many patients, such as obesity and insulin resistance, this risk increases up to 1400% (HR 14.13) [10]. Some authors distinguish between two different phenotypes of NAFLD: ‘genetic’ vs. ‘metabolic’. Therefore, metabolic NAFLD would be the phenotype associated with the development of T2DM, and its main characteristic is the presence of insulin resistance that precedes and originates the accumulation of lipids in the liver [11]. As this phenotype constitutes the most frequent clinical presentation, in 2020 and international expert panel from 22 countries proposed a novel definition of metabolic associated fatty liver disease (MAFLD) to replace “metabolic” NAFLD. Thus MAFLD is clinically defined by presence of obesity/overweight or T2DM or at least two metabolic risk abnormalities (hypertension, insulin-resistance, hyperlipidemia, CRP > 2 mg/L among other things) [12], however, understanding of the underlying mechanisms remains limited [13,14]. Insulin resistance has also been acknowledged as a MAFLD predictor in childhood, and both entities associate to an increase cardiometabolic burden [15,16], although the causal relationship among them seems more complex [17], as observed in young obese. In this context, the recent identification of new mediators that trigger inflammation, oxidative stress or lipotoxicity in the transition from steatosis to steatohepatitis, may be useful to prevent or retard the progression of the disease [18].

Genetic, chemical and dietary factors are the main inducers of MAFLD in preclinical models [19]. Among the most employed MAFLD genetic models are those derived from the ob/ob mutation (leptin deficiency), being C57BL/6J ob/ob mice the most widely known. Another ob/ob model, in which the leptin mutation is found in an insulin-resistant BTBR strain, is the BTBR ob/ob, leading to early T2DM development and maintenance of hyperglycemia, compared with other ob/ob models, as the animal become overweight/obese. This model is known due to its unique susceptibility to develop diabetic complications [20,21,22,23]. BTBR ob/ob strain background is a potent accelerator of kidney pathology, depicting histopathological lesions similar to those observed in human diabetic kidney disease [20]. Liver involvement in this model was also studied at early weeks and the authors concluded that the model was resistant to steatosis [24]. However, based on our observation that the livers of these animals did show indeed hepatomegaly at 12 weeks, we set out to characterize the hepatic involvement of the BTBR ob/ob mice, as well as, the pathophysiological mechanisms involved in MAFLD progression by using the BTBR ob/ob mouse model, in the presence/absence of obesity and hyperglycemia.

## 2. Results

### 2.1. Characterization of the Metabolic Factors Involved in Early MAFLD Progression in BTBR ob/ob Model

BTBR ob/ob mice cannot be differentiated phenotypically from their littermates BTBR WT until 4 to 6 weeks of life. Animals were fed with standard chow (used troughout the study) and changes in body weight and non-fasting glycemia were already detected from 4th and 6th week of life, respectively (Figure 1A,B). After 12 weeks of life, liver triglycerides content (Figure 1D), as well as serum metabolic parameters, were increased in BTBR ob/ob mice vs. BTBR WT, with significant changes in all studied variables (Table 1).

Macroscopical differences were observed in the liver of BTBR ob/ob mice during the first 12 weeks of life, mainly hepatomegaly (Figure 1C). Histopathologically, microvesicular steatosis was evident from week 6, progressing to macrovesicular during the following weeks (Figure 1F). At the 12th week of life, inflammatory clusters/foci and hepatocellular ballooning were also detectable (Figure 1G). 

### 2.2. Liver Inflammatory Cell Infiltration in the BTBR ob/ob Mice

The presence of inflammatory cells in the liver of BTBR ob/ob mice was analyzed at 12 weeks, including F4/80+ (monocytes/macrophages), CD3+ (T lymphocytes) and MPO+ (neutrophils). Even though no changes in F4/80+ cells between non-diabetic and diabetic mice were observed (Figure 2A,B), significant differences were detected in CD3+ T lymphocytes and MPO+ cells, meta-inflammation markers, observed in isolation at the interstitial level and forming part of inflammatory clusters (Figure 2C–F). 

### 2.3. Inflammatory, Oxidative Stress and Lipotoxic Markers in the Liver of BTBR ob/ob Mice

Intracellular signaling pathways related to inflammation, oxidative stress, and presence of lipotoxicity were activated at 12 weeks in our experimental model. At the immunohistochemical level, we observed increased JAK/STAT and NRF2 pathway activation (phosphorylation of the transcription factors STAT3 and NRF2, respectively) mainly in diabetic mice. Thus, pSTAT3 showed a clear cytoplasmic activation pattern, with minimal nuclear translocation in hepatocytes of the diabetic mice (Figure 3A,B). NRF2 showed a constitutive pattern in hepatocytes, with a notable nuclear translocation (Figure 3C,D). Lipid peroxidation was higher in livers from BTBR ob/ob mice, with increased expression of 4-hydroxynonenal (4-HNE), a marker of lipid peroxidation (Figure 3E,F). 

Gene expression studies confirmed the presence of oxidative stress in BTBR ob/ob mice, since we observed a marked upregulation of Nrf2 and its main effector HO-1 (Hmox-1), as well as downregulation of antioxidant enzymes such as Sod1 and Catalase. BTBR ob/ob mice showed a lower liver expression of the inflammatory cytokines Tnfα, Ifnγ and Il-12, but a significant increase of the chemokine Mcp-1 (Ccl2) (Figure 3G). Finally, scavenger receptors associated with fatty acid uptake (Cd36/Cd204) and efflux (Abca1/Abcg1) were also evaluated. An overexpression can be observed for Cd36 whereas Abca1 gene expression was downregulated. These results could indicate a possible intrahepatic lipid accumulation in BTBR ob/ob, favoring the uptake and limiting the efflux of fatty acids. 

### 2.4. Long-Term Metabolic and Morphological Changes Associated to MAFLD

To assess the metabolic and morphological changes associated to MAFLD, livers from non-obese and non-diabetic (BTBR WT), non-obese and diabetic (BTBR-STZ) and obese and diabetic (BTBR ob/ob) mice were evaluated in a later stage (22 weeks). This approach was performed to differentiate between body weight- and hyperglycemia-dependent liver changes. As reported in Table 2, BTBR ob/ob mice showed a marked difference in body weight, liver weight, food and water intake, glycemia, alkaline phosphatase (AP), triglycerides, total cholesterol and LDL.

Histopathological analysis disclosed manifestations of MAFLD in BTBR ob/ob mice, including micro and macrovesicular steatosis, predominantly in zone 2 and 3, inflammatory clusters with intrasinusoidal neutrophil infiltration, isolated megakaryoblasts, glycogenated nuclei and hepatocellular ballooning (Figure 4C 1–5). The steatosis showed a characteristic location by functional zones. In non-diabetic mice, lipid droplets were exclusively detected in zone 3. The STZ model showed mainly microvesicular steatosis in zone 2 and 3, and marginal macrovesicular. Obese mice have a predominant macrovesicular steatosis in zones 2 and 3. The NAFLD activity score demonstrated a higher activity grade in BTBR ob/ob mice (Figure 4D). Although Periodic acid-Schiff (PAS) staining is used for glycoproteins/glycolipids detection, is widely utilized to evaluate glycogen storage in liver. BTBR WT exhibited a symmetrical pattern of intracellular positive PAS staining. Nevertheless, an irregular arrangement was observed in both diabetic livers (Figure 4B). These changes were mainly evidenced in zone 3 or pericentral zone in the hepatocytes of STZ-treated mice, while the obese and diabetic BTBR ob/ob mice disclosed an accentuated location of steatosis in lipogenic zones, suggesting alterations of hepatocyte metabolism. Fibrosis was not observed (not shown).

### 2.5. Lipid Metabolism and Liver Lipidomics in the Experimental Model

BTBR ob/ob displayed a roughly 5-fold increase in total TG content in the liver as compared to BTBR WT, while when it was compared with STZ-treated BTBR mice this increase was even higher (Figure 5A). However, NEFA levels were only augmented in the liver and plasma of BTBR STZ mice (Figure 5B,C). To further explore the link between MAFLD and DNL, we analyzed the specific fatty acids that constitute the intrahepatic TG and NEFA fraction. In the hepatic TG fraction, focusing on those species which are the major products of DNL, BTBR ob/ob mice showed a significant increase in palmitic acid (C16:0; 562%), palmitoleic acid (C16:1n-7; 769%), stearic acid (C18:0; 130%), and oleic acid (C18:1n-9cis; 789%), compared to BTBR WT. These differences were even higher when comparing the results with the BTBR STZ mice (Figure 5D). This pattern was not observed for saturated fatty acids in NEFA fraction, since BTBR ob/ob mice showed a significant increase in palmitoleic (C16:1n-7; 104%), and oleic acid (C18:1n-9cis; 69%), compared with BTBR WT (Figure 5E). Detailed information about saturated fatty acids, mono-unsaturated fatty acids, and PUFA in TGs and NEFA fractions was described in Appendix A. 

The analysis based on the differences in fatty acids in the TG- and NEFA-fractions allowed us to discriminate each experimental models by using an Orthogonal Partial Least Squares Discriminant Analysis (oPLS-DA) (Figure 5F). In oPLS-DA, the component that best discriminated to the 3 different experimental groups was the presence of lipids associated with DNL (Appendix A). Furthermore, hierarchical clustering showed distinctive lipid profiles that associates with lipid composition in each one of the three groups (Figure 5G).

Different lipid ratios are used to assess the activity of the enzymatic pathways involved in their synthesis. Thus, DNL can be assessed by the ratio between palmitate (the main end product of DNL) and linoleate (whose origin is exclusively from the diet). The quantification of this ratio is 1.29 in BTBR WT animals, 2.67 in BTBR-STZ and 5.44 in BTBR ob/ob mice, with significant increases in the latter two vs. WT (*p* = 0.006 and *p* < 0.001, respectively). Similarly, the ratios between saturated and unsaturated FA showed significant decreases (BTBR WT: 3.56; BTBR-STZ: 2.16; BTBR ob/ob: 2.43, *p* = 0.01). Other elements measured were Elongase (oleate/palmitate); steaoryl-coA desaturase indices (palmitoleate/palmitate and oleate/stearate) in both cases were significantly increased in BTBR ob/ob vs. WT (0.11 vs. 0.16 *p* = 0.009) and 7.12 vs. 27.7 *p* < 0.0001). In contrast delta-(5)-desaturase (D5D) index (arachidonate/dihomo-γ-linolenate) significant variation was only observed in the STZ group (9.5 vs. 12.75 *p* = 0.001) [25].

### 2.6. Study of MAFLD Related Gene and Protein Expression in Liver from BTBR ob/ob Mice

In a next step, we analyzed the gene expression of makers associated with MAFLD progression. BTBR ob/ob mice showed a significant reduction in the main transcription factors related with lipo/glucogenic pathways, such as Srebp-1, Chrebp-1, Pparα, with significant increase of Pparγ activity (Figure 6A). These results were confirmed at the protein level by western blot analysis (Figure 6G,H). Moreover, we also observed a significant increase in gene expression of enzymes related with DNL (Acc1, Fasn, Scd1) in BTBR ob/ob mice, whilst Dgat2 enzyme, that catalyzed the final step in TG synthesis, was downregulated. The expression of scavenger receptors associated with fatty acids uptake (Cd36, Cd204) was also upregulated, in contrast with the downregulation of markers associated with efflux (Abca1, Abcg1) or intracellular transporters (Fatp2) (Figure 6B,C). Oxidative stress markers Nfr2 and HO-1 were augmented whereas a decrease on genes associated with mitochondrial biogenesis was evidenced (Figure 6D). Besides, most inflammatory chemokines measured were upregulated (Ccl2, Cxcl10, Cx3cl1) with lower expression of cytokines (IL-15) or innate receptors (TLR4) (Figure 6E). Also, pro-fibrotic markers as TGFβ, CTGF and Gremlin (Figure 6F) were downregulated.

## 3. Discussion

In this study, we aimed to describe the presence of MALFD in BTBR ob/ob mice and the potential pathogenetic mechanisms involved. This model was previously approached by Lan et al. while comparing microarray profile between two experimental models of diabesity: C57BL/6J (B6) ob/ob (diabetes resistant) and BTBR ob/ob (diabetes susceptible) [24]. Although we cannot compare our results with those of Lan et al. [24] as different controls were used, our data reaffirm the importance of the genetic background in the preclinical study of diabetes complications.

The comparative advantage of the C57BL/6J ob/ob model was the higher percentage of hepatic steatosis, but hyperglycemia was only present up to 14–16 weeks of life, and then returned to euglycemia. However, the BTBR strain was characterized by hyperglycemia that was maintained over time [26,27]. Leptin deficiency in BTBR strain is a key mechanism for the acceleration of hyperglycemia-associated damage, indicating the importance of the genetic background as responsible for the modification of the diabetic phenotype [28]. 

The discrepancy between leptin signaling dysfunction in murine models (ob/ob and db/db mice) and the hyperleptinemia detected in obese individuals is well known [29,30,31]. Leptin administration reduced hepatic steatosis in ob/ob mice by restoring adipose tissue and hepatic expression of aquaglyceroporins [32]. Leptin also alters energy intake and fat mass, but not energy expenditure in lean subjects [33]. In fact, reversibility of diabetic nephropathy in BTBR ob/ob mice was noted after recombinant leptin administration [20]. In this regard, it would be of interest to study the potential beneficial effect of leptin on the reversibility of hepatic steatosis in the BTBR strain. Therefore, the role of leptin in the progression of obesity complications needs to be further investigated. 

Hyperphagia observed in BTBR ob/ob leads to a 2-fold increased on daily food intake, body weight and liver weight compared to the non-diabetic control mice at 22 weeks. By clinical, analytical, and histopathological analysis, we have confirmed the presence of hepatic steatosis in early (12 weeks) and late (22 weeks) life stages of BTBR ob/ob mice. For that reason, we considered to further investigate the mechanisms involved in intrahepatic lipid accumulation, particularly whether it was merely a passive accumulation of dietary lipids or whether it was associated with DNL from carbohydrate sources and/or energy excess [34]. 

These mice show increased fatty acids uptake proteins and downregulation of efflux lipid transporters, both mechanisms promoting lipid accumulation in the hepatocyte. Of note, the activation of these pathways leads to the formation of monounsaturated fatty acids (C16:1n-7, C18:1n-9cis) and saturated fatty acids (C16:0 and C18:0), associated with DNL. We observed that BTBR ob/ob mice depicted abnormalities in the receptors related to influx and efflux of fatty acids, as well as on the activation of the mechanisms involved in mitochondrial dysfunction and DNL. The synergy of these factors leads to an increased ectopic accumulation of fatty acids in the liver. Additionally, in order to evaluate the effect of hyperglycemia on in liver TG and NEFA in euglycemic normal weight mice (BTBR WT), hyperglycemic/normal weight (BTBR + STZ) and hyperglycemic mice and obese mice (BTBR ob/ob).

Lipidomic analysis of intrahepatic TG and NEFA fractions showed a distinctive profile in each experimental group. Since the diet was identical in all groups, these results may suggest the existence of a different metabolic lipid synthesis rate. Of interest, the lipid profile associated with BTBR ob/ob mice showed an enrichment in de novo synthesized lipids (C16:0, C18:0, C16:1n-7 and C18:1n-9cis), a picture similar to that observed in subjects with MAFLD [35]. Meanwhile in BTBR-STZ mice there was an increase in the NEFA subfraction and a trend toward reduction of fatty acids accumulation in the triglycerides fraction. Several authors have demonstrated the inhibitory effect of NEFAs on glucose metabolism (transport, phosphorylation and oxidation), mainly by impairing insulin signaling in the muscle [36,37,38,39]. 

Indeed, the results obtained in this study support the idea that obesity, in a hyperglycemic context, activates intracellular signaling pathways in the liver related to the synthesis of new fatty acids. This effect, dependent on daily intake and not of diet type, may be one of the mechanisms responsible of the progression of liver steatosis in BTBR ob/ob model. 

Consumption of a high-fat diet or diet-induced obesity (DIO) increases body weight, insulin resistance and macrovesicular steatosis; however, these preclinical models do not produce significant hyperglycemia. These facts demonstrate that intrahepatic lipid accumulation is caused by both liver and adipose tissue dysfunction, an effect that may be aggravated when insulin resistance and hyperglycemia are permanent [40]. In clinical practice, it is impossible to separate lipo/glycogenic factors involved in the MAFLD progression, however, the modulation of DNL could be a good therapeutic option. For that reason, we studied the metabolic pathways altered in this pathological scenario. A reduction in the main regulators of lipid synthesis (ChREBP1 and SREBP1c) was observed, whereas PPARγ was overexpressed, suggesting an increase in lipid storage, as previously described [41,42,43]. In addition, the increased expression of enzymes related to fatty acid synthesis (ACC1, FASN and SCD-1) confirmed the data obtained by mass spectrometric analysis.

In conclusion, BTBR ob/ob mice strain constitutes an excellent experimental model of MAFLD, showing not only liver steatosis, but also recruitment of inflammatory cells, activation of inflammatory signaling pathways, oxidative stress and lipotoxicity, all them described as meta-inflammation [44]. These features are associated with the transition from steatosis to steatohepatitis, which characterize the pathological progression of MAFLD. This experimental model resembles early stages of human MAFLD and may be an excellent translational model between MAFLD, T2DM and diabetic complications [20,45,46,47]. Therefore, therapeutic strategies to limit cell damage and ectopic lipid accumulation, through the selective modulation of these particular intracellular signaling pathways in early stages of MAFLD, could be promising.

## 4. Materials and Methods

### 4.1. Experimental Model

In this study we studied MAFLD progression on 1) non-diabetic control mice (Black Tan and Brachyuric (BTBR WT mice), 2) diabetic and non-obese mice (BTBR mice treated with low doses streptozotocin, BTBR-STZ), 3) diabetic and obese mice (BTBR and ob/ob (leptin deficient) (BTBR.Cg-Lepob/WiscJ; RRID:IMSR_JAX:004824). BTBR WT and BTBR ob/ob male mice were sacrificed at 4, 6, 12 and 22 weeks (*n* = 5–6 mice/group). Standard chow diet (LASQCdiet Rod14-H) with a low-fat content (3.5%) and water ad libitum were available. Daily intake (water and food) was measured. In parallel, 12-week-old BTBR WT mice were intraperitoneally injected with low-doses of STZ (50 mg/kg) for 5 consecutive days and 10% sucrose water was supplemented to avoid hypoglycemia post-streptozotocin injection, according to the recommendations of the Animal Models Diabetic Complications Consortium (DiaComp). All mice included in BTBR-STZ group manifested glycemia > 300 mg/dl for 9–10 weeks until sacrifice (22 weeks-old). All mice included in BTBR-STZ group manifested glycemia > 300 mg/dl for 9–10 weeks until sacrifice (22 weeks-old) (Appendix A).

The measurement of glycemia and body weight was made every week using a glucometer NovaPro (Nova Biomedical Iberia, Barcelona, Spain) and digital balance, respectively. Animals were euthanized under anesthesia (ketamine 100 mg/kg and xylazine 10 mg/kg). Post-anesthetic assessment, liver and blood sample for serum collection were taken. Breeding pairs BTBR heterozygotes (BTBR ob+/−) were purchased from Jackson Laboratories (Bar Harbor, ME, USA) and the colony was expanded in-house. Animals were housed at a density of four animals per cage in a controlled environment in individually ventilated cages (20–22 °C) with 12-h light–dark cycles. 

In this study, there were no exclusions and no randomization was performed due to phenotypic differences between groups. 

### 4.2. Biochemical Parameters

Lipid profile and liver biochemical parameters were assessed in serum, including triglycerides (TGs), total cholesterol, high-density lipoprotein cholesterol (HDL), and low-density lipoprotein cholesterol (LDL) by Friedwald formula, alanine transaminase (ALT), aspartate transaminase (AST), alkaline phosphatase (AP) and albumin. Serum was collected from femoral artery under anesthesia prior to animal sacrifice into a Vacutainer ACD blood collection tubes (Becton Dickinson and Company, Plymouth, UK). These measurements were performed in a Roche Cobas autoanalyzer’s at the central laboratories of our Institution. Liver lipids were extracted using Folch extraction (chloroform–methanol) [48], and NEFA levels, both in serum and liver extract, were measured using NEFA C enzymatic assay kit (WAKO, Neuss, Germany) as described [49]. Liver TGs content was measured using the GPO-trinder method, for which the TG colorimetric assay kit was employed following the manufacture’s instructions (Cayman Chemical Company; Ann Arbor, MI, USA). 

### 4.3. Lipid Profile Determination

Fatty acid methyl esters (FAMEs), TGs and non-esterified fatty acids (NEFAs) from liver were determined as follows. A weighed amount of liver from 22 weeks mice (around 20 mg) was placed in a borosilicate glass tube (previously washed with n-hexane) containing 1 mL of 0.9% sterile sodium chloride. Tissue was homogenized on ice in two 10-s series with an OMNI TH Tissue Homogenizer (Omni International, Kennesaw, GA, USA). The homogenate was spiked with 100 μL of the internal standard (ISTDs) trinonadecanoin and nonadecanoic acid (100 μg/mL solution (10 μg) each; Nu-Chek Prep, Elysian, MN, USA) and the lipids were extracted with 2 mL of chloroform/methanol (2:1 *v*/*v*). After centrifugation (5 min at 3500 rpm), the organic phase was transferred to a new borosilicate glass tube and evaporated to dryness under N2 at 30 °C. TGs and NEFAs were isolated by solid-phase extraction as described in Burdge et al. [50]. Fatty acids were hydrolyzed and methylated following an adaptation of the method described by Agren et al. [51]. Briefly, 100 μL of n-toluene and 500 μL of boron trifluoride-methanol solution (14%) were added to the tube, which was capped and placed into a block heater (100 °C) for 60 min. After cooling, 500 μL of distilled water and 500 μL of n-hexane were added. The tubes were shaken for 1 min and then centrifuged for 5 min at 3500 rpm at room temperature to separate the layers. The hexane layer was placed in a test tube and evaporated to dryness under N2 at 30 °C. The extracts were reconstituted with 100 μL of n-hexane and transferred to an automatic injector vial containing a glass insert of 300 μL.

FAMEs were analyzed by gas chromatography/electron ionization mass spectrometry (GC/MSEI), using an Agilent 6890N GC equipped with an Agilent 7683 autosampler, and an Agilent 5973N mass spectrometry detector. FAMEs were separated with a J&W DB-Fast FAME capillary column (30 m × 0.2 mm × 0.25 μm film thickness) (Agilent Technologies; Santa Clara, CA, USA). The injector temperature was set at 250 °C, and 1 μL injections were performed (split ratio 25:1). GC was run using an optimized temperature program, as follows: the program started at 50 °C, held for 0.5 min, increased to 194 °C at a rate of 25 °C/min, held for 1 min, increased to 245 °C at a rate of 5 °C/min, and held for 3 min. Helium was used as a carrier gas (14 psi, constant pressure mode). FAMEs were detected using the selected ion monitoring (SIM) mode. Based on the work of Thurnhofer and Vetter [52], several m/z ions common to saturated, monounsaturated, and polyunsaturated FAMEs were monitored. Twelve mixtures of FAME external calibration standards were prepared by diluting FAME mix certified reference material (Supelco 37 Component FAME Mix, Merck) in hexane. These standards were kept at −80 °C until analysis. 40 μL of each mixture were added to a tube, were spiked with 100 μL of the ISTD C19:0-methyl ester (100 μg/mL solution (10 μg)), evaporated to dryness under N2 at 30 °C, reconstituted with 100 μL of hexane, and transferred to an automatic injector vial containing a glass insert of 300 μL. The equivalents of C19:0 added to the samples as TG and NEFA ISTDs were the same as the amount of C19:0-methyl ester added to the external calibrators. The concentration of FAMEs in the samples were calculated by linear regression of the peak area ratio relative to that of the internal standard. The normalized concentrations were calculated by dividing the concentrations by the weight of the liver tissue.

### 4.4. Histological Analysis and Immunohistochemistry Studies

A section of liver was fixed in 4% formaldehyde and further embedded in paraffin. 4 µm tissue sections were stained for histochemical studies (H&E/Periodic Acid Schiff) and immunohistochemistry. The liver sections were classified according to a semiquantitative histopathological score damage (NAS, NAFLD activity score) performed by a trained personnel in a blinded manner [53]. The primary antibodies for immunodetection were: F4/80 monocytes/macrophages ([1:70], MCA497,RRID:AB_2098196, Bio-Rad; Hercules, CA, USA), CD3 T lymphocytes ([1:100], M7254, RRID:AB_2631163, Agilent Technologies; Santa Clara, CA, USA), MPO ([1:100], TC66701, RRID:AB_579628, Agilent Technologies; Santa Clara, CA, USA), phosphorylated (p-) STAT3 serine 727 ([1:100], 9134, RRID:AB_331589, Cell Signaling Technology; Danvers, MA, USA), p-nuclear factor erythroid 2-related factor 2 (NRF2) serine 40 ([1:2000], ab76026, RRID:AB_1524049, Abcam; Cambridge, UK) and 4-hydroxy-2-nonenal (4-HNE) ([1:200], ab46545, RRID:AB_722490, Abcam; Cambridge, UK). All primary antibodies were assessed by indirect immunoperoxidase ([1:2000], except for p-NRF2, which incubated with the Vectastain Elite ABC HRP Kit RTU (PK-7100, RRID:AB_2336827, Vector Laboratories; Burlingame, CA, USA). Sections were revealed with ImmPACT DAB Peroxidase Substrate (SK-4105, RRID:AB_2336520, Vector Laboratories; Burlingame, CA, USA) and counterstained with hematoxylin (Thermo Fisher Scientific; Waltham, MA, USA). Positive staining was quantified using Image-Pro Plus software and expressed as percentage of the total area or number of positive cells (per 10 random fields).

### 4.5. Gene Expression Studies

Total RNA from liver tissue was isolated with TRIdity G A4051 (Panreac AppliChem, Barcelona, Spain). Complementary DNA (cDNA) was synthesized by a High-Capacity cDNA Archive Kit (Applied Biosystems, Foster City, CA, USA) using 2 µg total RNA primed with random primers following the manufacturer’s instructions. Quantitative gene expression analysis was performed by qRT-PCR (Quantitative real-time PCR 7500 Applied Biosystems, System SDS software V.1.2b1c3) using TaqMan gene expression assays for mouse and primers designed through Primer-BLAST software and synthesized by Thermo Fisher Scientific; Waltham, MA, USA. The expression of targets genes was analyzed in duplicate and normalized to housekeeping gene 18s rRNA. Gene expression changes are referred versus the average gene expression in BTBR WT animals (normalized as 1), therefore each gene is shown as fold change. The primers for PCR detection are listed in Appendix A.

### 4.6. Protein Studies

Liver tissue samples were homogenized in lysis buffer (50 mM Tris–HCl, 150 mM of NaCl, 2 mM of EDTA, 2 mM of EGTA, 0.2% Triton X-100, 0.3% Igepal complemented with protease inhibitor cocktail (CP8340, Sigma-Aldrich; Saint Louis, MO, USA) and phosphatase inhibitor cocktail (P0044, Sigma-Aldrich; Saint Louis, MO, USA) and quantified using the BCA protein assay kit (Thermo Fisher Scientific; Waltham, MA, USA) to later separate the proteins (50 μg) in 8–12% acrylamide gels using SDS-PAGE. After electrophoresis, samples were transferred to PVDF membranes (IPVH00010, Millipore, Bedford, MA, USA), blocked in TBS containing 0.1% Tween 20 (TBS-T) and 5% skimmed milk for 1 h at room temperature and incubated in the same buffer with different primary antibodies overnight at 4 °C. The following primary antibodies were employed: SREBP1 ([1:500], NB600–582, RRID:AB_10001575, Novusbio, Bio-Techne; Minneapolis, MN, USA), PPARγ (E-8) ([1:500], sc7273, RRID:AB_628115, Santa Cruz Biotechnology; Santa Cruz, CA, USA), DGAT2 ([1:1000], ab237613, Abcam; Cambridge, UK), SCD1 (C12H5) ([1:1000, #2794S, RRID:AB_2183099, Cell Signaling Technology; Danvers, MA, USA) and FASN (C20G5) (1:1000, #3180S, RRID:AB_2100796, Cell Signaling Technology; Danvers, MA, USA). After that, blots were washed with TBST and incubated 1 h at room temperature with the appropriate HRP (horseradish peroxidase)-conjugated secondary antibody (anti-mouse or anti-rabbit, 1:2000 dilution, Invitrogen; Waltham, MA, USA). After being washed with TBST, blots were developed with the chemiluminescence method (ECL Luminata Crescendo, WBLUR0500, Millipore; Burlington, MA, USA) and scanned using the ImageQuant LAS-4000 (GE Healthcare; Chicago, IL, USA). Blots were then probed with mouse monoclonal anti-β-Actin antibody (1:5000 dilution, A2228, Sigma-Aldrich; Saint Louis, MO, USA) and levels of expression were corrected for minor differences in loading. Results were quantified using Quantity One software (Bio-Rad; Hercules, CA, USA) and expressed as densitometric arbitrary units (AU). 

### 4.7. Statistical Analysis

The data are presented as scatter dot plots with mean ± SEM (graphs) or median ± IQR (tables) of the total number of animals. Graphs and corresponding statistical tests were carried out in R (v4.0.2) or GraphPad Prism V.6 software (GraphPad Software Inc., La Joya, CA, USA). Statistical analyses were performed using Student’s t or non-parametric Mann-Whitney U test for comparison between two groups. Lipid discriminant and clustering analysis was carried out using respectively an Orthogonal Partial Least Squares Discriminant Analysis (oPLS-DA) and a Euclidean Hierarchical clustering with the MetaboAnalystR 2.0 package [53]. Differences were considered statistically significant at *p* < 0.05.

## Figures and Tables

**Figure 1 ijms-23-03965-f001:**
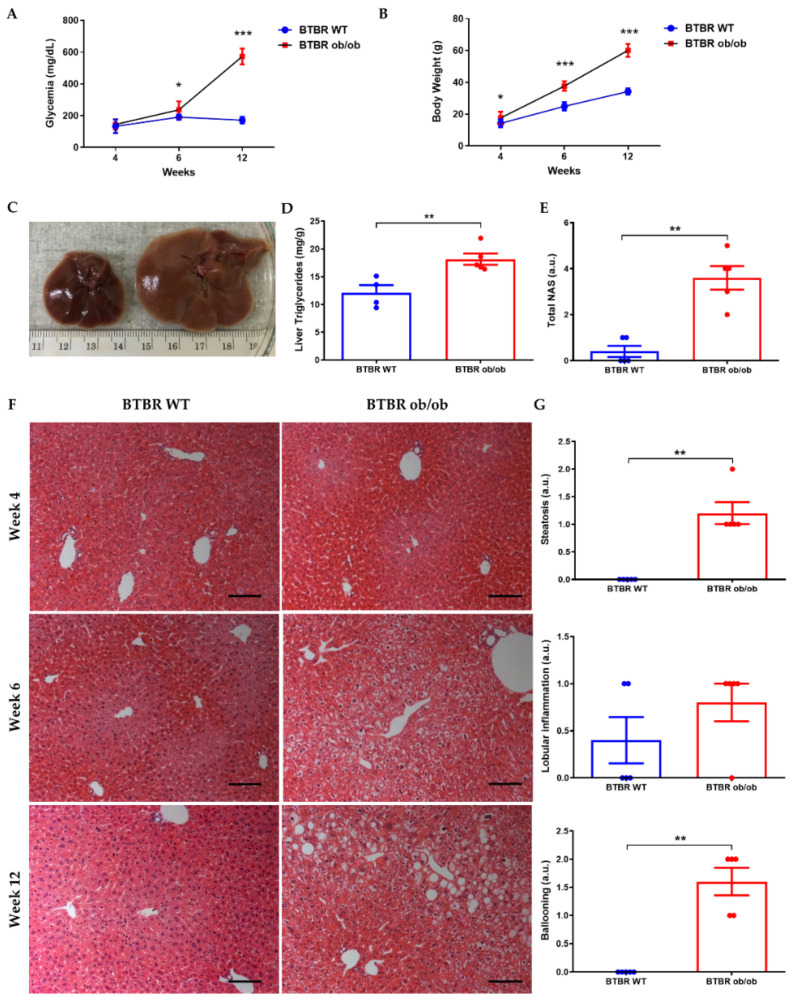
**Progression of liver steatosis from weeks 4 to 12.** Non-fasting glycemia (**A**) and body weight (**B**) of BTBR WT and BTBR ob/ob mice at 4, 6 and 12 weeks. (**C**) Macroscopic photograph of liver at 12 weeks in BTBR WT (left) and BTBR ob/ob mice (right). (**D**) Measurement of liver TGs at 12 weeks. (**E**) NAFLD activity score (NAS) assessment at 12 weeks (**F**) Representative H-E images of livers of BTBR WT (left) and BTBR ob/ob mice (right) showing steatosis progression (magnification 100×) from 4–12 weeks. (**G**) Semi-quantitative determination of steatosis, lobular inflammation and hepatocytes ballooning in livers of mice at 12 weeks. Data are shown as scatter dot plots and mean ± SEM of each group (*n* = 5–6 mice/group); * *p* < 0.05, ** *p* < 0.01, *** *p* < 0.001 vs. BTBR WT. Abbreviations: NAS: NAFLD activity score; a.u: arbitrary units.

**Figure 2 ijms-23-03965-f002:**
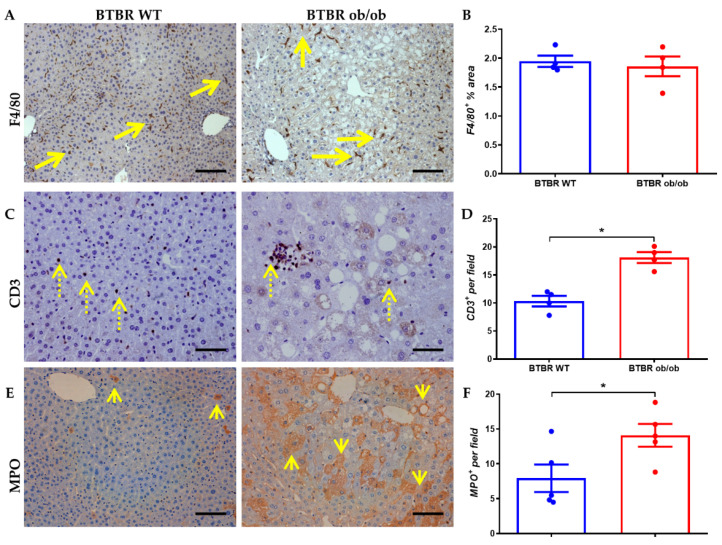
**Inflammatory infiltrate in BTBR WT (left) and BTBR ob/ob mice (right) at 12 weeks.** (**A**) Representative image of monocytes/macrophages infiltration (F4/80+ staining; long arrow) (magnification 100×), and F4/80+ area quantification (**B**). (**C**) Representative image (of lymphocytes T (CD3; dotted arrow) staining (magnification 200×) and its quantification (**D**). (**E**) Myeloperoxidase (MPO; short arrow) immunostaining (magnification 200×) and quantification of the positive cells per field (**F**). Data are shown as scatter dot plots and mean ± SEM of each group (*n* = 5 mice/group); * *p* < 0.05 vs. BTBR WT.

**Figure 3 ijms-23-03965-f003:**
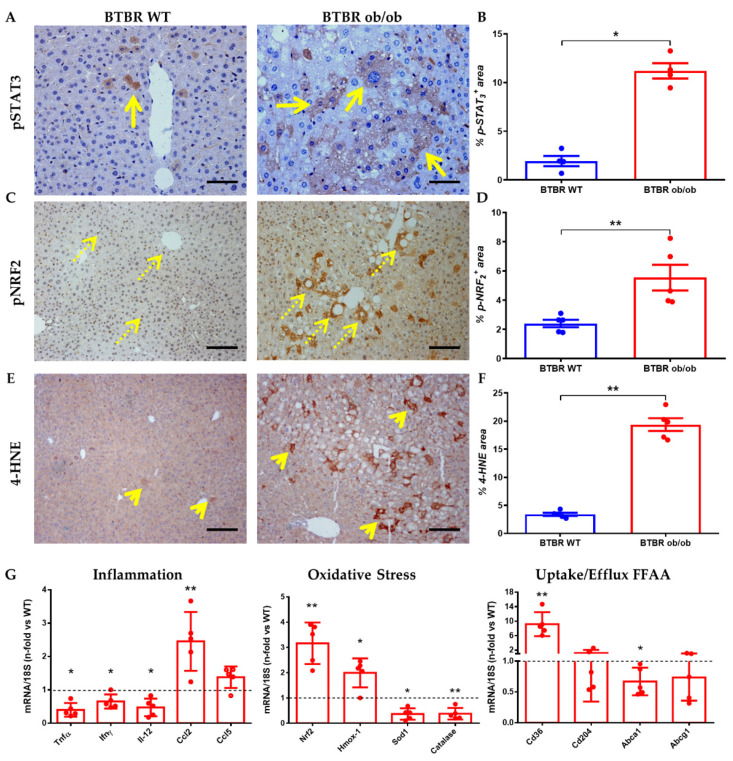
**Comparison of inflammatory, oxidative stress and lipid markers between BTBR WT (left) and BTBR ob/ob (right) at 12 weeks**. (**A**) Representative image of pSTAT3 (long arrow) immunostaining (magnification 200×) and (**B**) pSTAT3 activation quantification. (**C**) pNRF2 staining (dotted arrow) as oxidative stress response (magnification 100×) and area quantification (**D**). (**E**) 4-HNE staining (short arrow) as lipoperoxidation (magnification 100×) and its quantification (**F**). In the lower row (**G**) is shown mRNA expression of several inflammation (Tnfα, Ifnγ, Il-12, Ccl2 and Ccl5); oxidative stress (Nrf2, Hmox-1, Sod-1 and Catalase) and lipid uptake/efflux (CD36, CD204, ABCa1 and ABCg1) genes. Data are shown as scatter dot plots and mean ± SEM of each group (*n* = 5 mice/group); * *p* < 0.05, ** *p* < 0.01, vs. BTBR WT.

**Figure 4 ijms-23-03965-f004:**
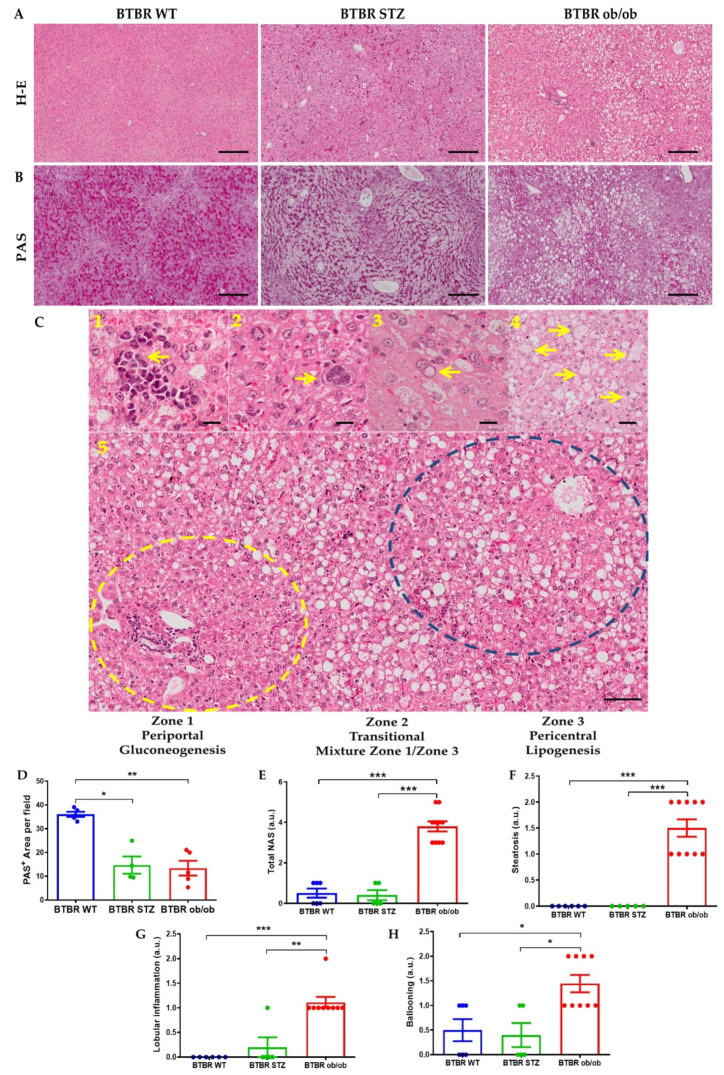
**Liver histopathological changes at 22 weeks between BTBR WT, BTBR-STZ and BTBR ob/ob.** Representative images of H-E (**A**) and PAS (**B**) staining in the 3 animal groups. (**C**) Major findings associated to MAFLD development: 1. Inflammatory clusters; 2. Isolated megakaryoblast; 3. Glycogenated nuclei; 4. Hepatocellular ballooning. 5. Steatosis distribution into 3 distinctive liver zones. (**D**) Quantification of positive Periodic Acid Schiff (PAS) staining, as glycogen liver deposition in BTBR WT, STZ and ob/ob mice. (**E**) Quantification of NAFLD activity score (total NAS) and its histopathological characteristics: (**F**) steatosis, (**G**) lobular inflammation and (**H**) hepatocytes ballooning. Data are shown as scatter dot plots and mean ± SEM of each group (*n* = 5–8 mice/group); * *p* < 0.05, ** *p* < 0.01, *** *p* < 0.001 vs. BTBR WT or BTBR-STZ.

**Figure 5 ijms-23-03965-f005:**
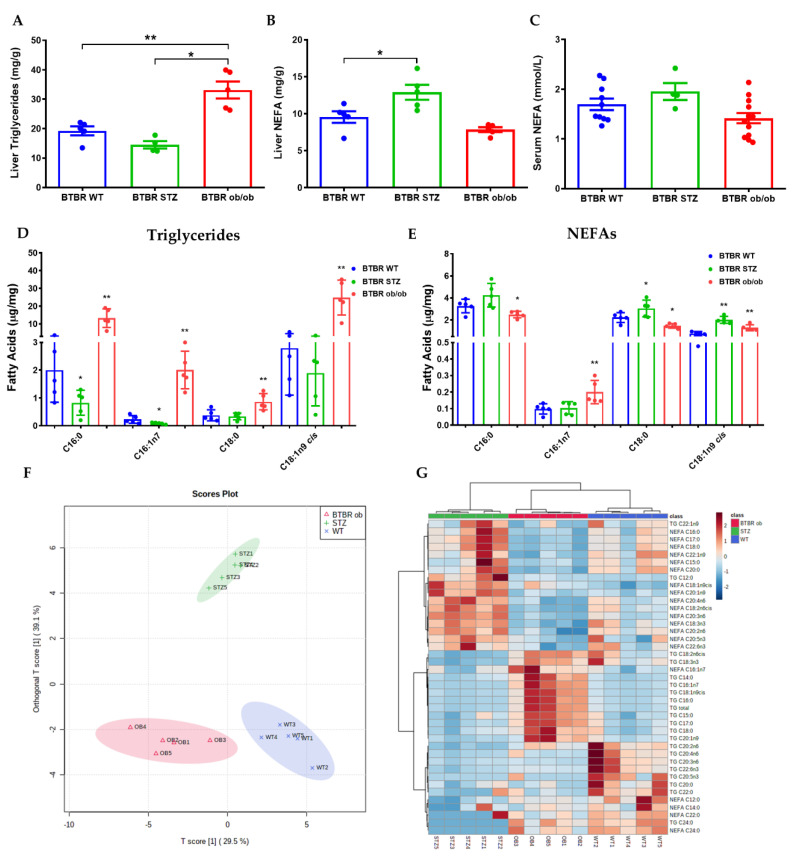
**Lipid composition in liver at 22 weeks from BTBR WT (blue), BTBR−STZ (green) and BTBR ob/ob mice (red).** Measurement of liver TGs (**A**), liver NEFAs (**B**) and serum NEFAs (**C**). Analysis by GC−MS of fatty acids, involved in de novo lipogenesis, from triglycerides (**D**) and NEFAs (**E**). (**F**) Orthogonal Partial Least Squares Discriminant Analysis (oPLS−DA) based on liver lipid composition data showing a clear separation between models. (**G**) Hierarchical clustering based on liver lipid composition showing the upregulated lipids (dark brown) and downregulated (dark blue) in each experimental group. Data are shown as scatter dot plots and mean ± SEM of each group (*n* = 5–8 mice/group); * *p* < 0.05, ** *p* < 0.01 vs. BTBR WT or BTBR−STZ.

**Figure 6 ijms-23-03965-f006:**
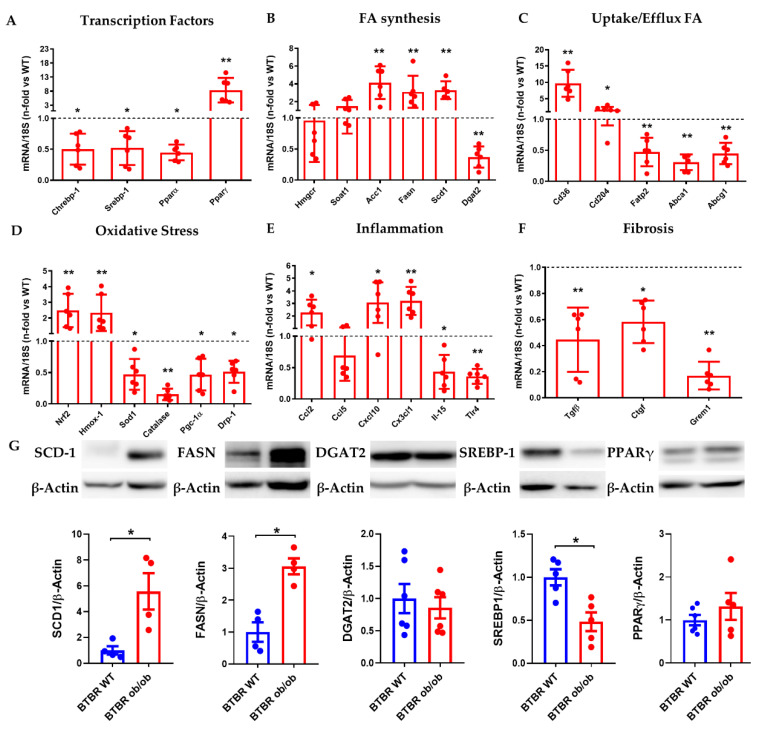
**Liver gene and protein expression at 22 weeks.** (**A**) mRNA expression by RT-qPCR of main transcription factors (Chrebp-1, Srebp-1, Pparα, Pparγ), (**B**) fatty acids synthesis (Hmgcr, Soat1, Acc1, Fasn, Scd1, Dgat2), (**C**) uptake/efflux fatty acids (Cd36, Cd204, Fatp2, Abca1, Abcg1), (**D**) oxidative stress (Nrf2, Hmox-1, Sod1, Catalase, Pgc-1α, Drp-1), (**E**) Inflammation (Ccl2, Ccl5, Cxcl10, Cx3cl1, Il-15, Tlr4), and (**F**) fibrosis (Tgfβ, Ctgf, Grem1) was evaluated. Fold changes of the target gene was normalized by of their respective housekeeping gene 18s ribosomal subunit, (**G**) Protein expression of lipogenic enzymes by western blot was evaluated. Fold changes of proteins levels in BTBR ob/ob vs. BTBR WT (n-fold = 1) normalized by β-Actin and images of their respective Western Blot. Data are shown as scatter dot plots and mean ± SEM of each group (*n* = 6 mice/group); * *p* < 0.05, ** *p* < 0.01 vs. BTBR WT.

**Table 1 ijms-23-03965-t001:** **Metabolic and biochemical parameters in BTBR WT and BTBR ob/ob at 12 w.** Data are shown as Median (IQR). ** *p* < 0.01; *** *p* < 0.001 vs. BTBR WT. Abbreviations: AST: aspartate aminotransferase; ALT: alanine aminotransferase; AP: Alkaline Phosphatase; HDL-C: high-density lipoprotein cholesterol LDL-C: low-density lipoprotein cholesterol. ^&^ Undetectable. ^%^ n-fold change vs. BTBR WT.

Variables	BTBR WT	BTBR ob/ob	n-Fold Change ^%^
**AST (IU/L)**	**46** (39.75, 50.75)	**76** (66.5, 58) **	**1.65**
**ALT (IU/L)**	**27** (23, 30.25)	**6****2** (52, 72.5) ***	**2.29**
**AST/ALT ratio**	**1.66** (1.39, 2.16)	**1.09** (1.02, 1.46)	**-**
**AP (IU/L)**	**60.5** (54.75, 84.25)	**146** (123.5, 163.5) ***	**2.41**
**Total Cholesterol (mg/dL)**	**118** (103.3, 126.8)	**215** (181, 275) **	**1.82**
**Triglycerides (mg/dL)**	**123.5** (99.25, 129)	**253.5** (192, 310.5) **	**2.05**
**HDL (mg/dL)**	**101** (89.75, 109.3)	**149** (141, 198) **	**1.48**
**LDL (mg/dL)**	**0** (0, 0) ^&^	**19** (4, 26,5)	**-**

**Table 2 ijms-23-03965-t002:** **Metabolic and biochemical parameters in BTBR WT, BTBR-STZ and BTBR ob/ob at 22 weeks.** Data are shown as Median (IQR). * *p* < 0.05; ** *p* < 0.01; *** *p* < 0.001 vs. BTBR WT § non-parametric data: Mann-Whitney U test. Abbreviations: LW/BW: Liver weight/body weight ratio; AST: aspartate aminotransferase; ALT: alanine aminotransferase; AP: Alkaline Phosphatase; HDL-C: high-density lipoprotein cholesterol LDL-C: low-density lipoprotein cholesterol. ^&^ Undetectable. ^%^ n-fold change BTBR ob/ob vs. BTBR WT.

Variables	BTBR WT	BTBR STZ	BTBR ob/ob	n-Fold Change ^%^
**Water intake (mL/day)**	**5** (5, 6)	-	**38** (32, 45) ***^, §^	**7.60**
**Food intake (g/day)**	**6.7** (6.3, 7.8)	-	**13.12** (12.17, 14.13) ***^, §^	**1.96**
**Liver weight (g)**	**2.70** (2.60, 3.20)	-	**6.19** (5.90, 6.82) ***	**2.29**
**Body weight (g)**	**37** (36, 38)	**31** (30, 34) ***^, §^	**71** (69, 72) ***	**1.91**
**LW/BW ratio**	**2.7** (2.5 3.2)	-	**6.2** (5.9 6.9) *	**2.30**
**Glycemia 22 week (mg/dL)**	**152** (149, 157)	**350** (265, 380) ***^, §^	**525** (493, 569) ***	**3.45**
**AST (IU/L)**	**80** (64, 134)	**88** (83, 95)	**84** (74, 116) ^§^	**1.05**
**ALT (IU/L)**	**14** (11, 30)	**38** (35, 42)	**40** (35, 52) **^, §^	**2.86**
**AST/ALT ratio**	**5.3** (3.9, 6.9)	**2.0** (2.5 2.7)	**2.0** (1.7, 2.7) ***	**-**
**AP (IU/L)**	**38** (34, 54)	**127** (108, 141)	**97** (74, 113) ***	**2.55**
**Total Cholesterol (mg/dL)**	**128** (121, 142)	**112** (104, 120) **^, §^	**204** (166, 233) ***	**1.59**
**Triglycerides (mg/dL)**	**56** (53, 73)	**134** (110, 151)	**108** (98, 136) ***	**1.93**
**HDL (mg/dL)**	**102** (98, 108)	**96** (88, 103) **^, §^	**158** (122, 175) ***	**1.55**
**LDL (mg/dL)**	**10** (7, 27)	**0** (0, 0) ^&^	**29** (24, 38) *^, §^	**2.90**

## Data Availability

The database of the measurements and analytics will be available on request.

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
