# Peer review of "Meta-Inflammation and De Novo Lipogenesis Markers Are Involved in Metabolic Associated Fatty Liver Disease Progression in BTBR ob/ob Mice"

_ijms, 2022, doi:10.3390/ijms23073965_

Round 1

Reviewer 1 Report

In their study Opazo-Rios et al. describe the development of MAFLD in the combined genetic mouse model of BTBR ob/ob mice. They carefully describe the development of liver pathology and provide evidence of an enhance de novo fatty synthesis in BTBR ob/ob mice compared to BTBR mice. Importantly, they compare their results to a previous study by Lan et al (Diabetes 52:688–700, 2003) in which BTBR ob/ob mice were compared to ob/ob mice. 

Although the current study is very interesting, it falls short of providing essential data to support the hypothesis put forward. In addition, it misinterprets the data of the previous study by Lan et al. and therefore draws ill conclusions.  The important points are as follows:

Main: The current study does NOT include the ob/ob strain in the experiment. While it is evident from the current experiments that the ob/ob background in the BTBR ob/ob strain does indeed favor MAFLD development when compared to the BTBR strain, the authors fail to provide a direct comparison with the ob/ob strain. This direct comparison is essential for their conclusion and would, indeed be very interesting. 

Suggestion: It is certainly difficult to add these experiments post hoc. However, the ob/ob strain is a widely studied strain and the authors might try if they can retrieve the relevant data from a broad array of publicly available data sets. 
A comparison of the expression levels with the study of Lan et al. is not possible, because Lan et al. normalized all their data to ob/ob. The direct comparison of the absolute values of triglyceride levels in the livers at 12 resp. 14 weeks already indicates that the phenotypes might not differ, but just the aim of the study, in one case to prove the development of MAFLD, in the other to prove that ample storage of triglycerides in the liver might protect from β-cell damage.

Minor:
The manuscript is poorly prepared and needs thorough editing. Examples (not comprehensive):

Line 57 "proposed" not propose

Line 75 "Also" not Although

Line 80, remove wild card text

Results 2.1 a brief remark should be included that only one type of diet is used. 

Table 1: Headings it is not WT and ob/ob but BTBR and BTBR ob/ob

Line 97 line feeds after table are missing 

Fig. 1. To allow a direct comparison with ref. 21, the liver TG content as mg/g should be provided

Fig. 3. G provide exact explanation how the wild type "1" was determined.

Table 2  it would be useful to determine the glycogen content  in mg/g

Line 170, again line feed after table is missing

Discussion line 250 and subsequent: see above, the discussion of the results of the Lan publication (ref. 21) is completely misleading.

Line 291, the sentence does not make sense. 

Fig. 7 nice, but absolutely unnecessary and not providing any functional connections.   

The discussion is rather long and ought to be shortened.

Methods, line 370  the conditions under which the animals were kept prior to killing (8 h fast, 4 h fast???) are missing. 

4.4 it should be stated, whether the person who did the scoring was blinded for the experimental group. 

Author Response

Dear reviewer

We thank you in advance for your feedback.

Reviewer 1

In their study Opazo-Rios et al. describe the development of MAFLD in the combined genetic mouse model of BTBR ob/ob mice. They carefully describe the development of liver pathology and provide evidence of an enhance de novo fatty synthesis in BTBR ob/ob mice compared to BTBR mice. Importantly, they compare their results to a previous study by Lan et al (Diabetes 52:688–700, 2003) in which BTBR ob/ob mice were compared to ob/ob mice.

Although the current study is very interesting, it falls short of providing essential data to support the hypothesis put forward. In addition, it misinterprets the data of the previous study by Lan et al. and therefore draws ill conclusions.  The important points are as follows:

Main: The current study does NOT include the ob/ob strain in the experiment. While it is evident from the current experiments that the ob/ob background in the BTBR ob/ob strain does indeed favor MAFLD development when compared to the BTBR strain, the authors fail to provide a direct comparison with the ob/ob strain. This direct comparison is essential for their conclusion and would, indeed be very interesting.

Suggestion: It is certainly difficult to add these experiments post hoc. However, the ob/ob strain is a widely studied strain and the authors might try if they can retrieve the relevant data from a broad array of publicly available data sets.

A comparison of the expression levels with the study of Lan et al. is not possible, because Lan et al. normalized all their data to ob/ob. The direct comparison of the absolute values of triglyceride levels in the livers at 12 resp. 14 weeks already indicates that the phenotypes might not differ, but just the aim of the study, in one case to prove the development of MAFLD, in the other to prove that ample storage of triglycerides in the liver might protect from β-cell damage.

General comments.

Firstly, we would like to express our sincere appreciation for the reviewer comments and suggestions, which will surely contribute to improve the quality of the manuscript. The specific answers for each comment are as follows:

Although leptin deficiency (ob/ob) in the C57BL/6 strain is much more “effective" in favoring the development a greater amount of steatosis and therefore hepatic triglyceride content than the BTBR strain early in life, its use as a preclinical tool is strongly limited by the normalization of glycemia at 14 weeks of life, as it does not recapitulate to what happens in diabetic or obese patients with NAFLD/MAFLD.

As mentioned, our group has been studying this animal model for years, both at the cardiovascular, renal and retinal throughout its life span and, similarly to several authors, we believe that the BTBR ob/ob mice represents an excellent animal model for the study of the complications of DM (20634301; 24944269; 30094465; 32545818)

On the whole, we believe that BTBR ob/ob mice constitutes an excellent experimental model of MAFLD, showing not only liver steatosis, but also recruitment of inflammatory cells, activation of inflammatory signaling pathways, oxidative stress and lipotoxicity, features associated with the transition from steatosis to steatohepatitis, as seen in the progression of MAFLD.

Triglycerides were determined in 5 BTBR and 5 BTBR ob/ob animals at 12 weeks of age whose livers were frozen, obtaining an amount of 13-16.5 mg/g in BTBR and 36-77 mg/g in BTBR ob/ob, similar results obtained for Lan et al.

The problem is that, in order to make an accurate comparative measurement, we think it would be necessary to do the quantifications at the same time, which would require additional time to complete the study. So in case the referee considers this data essential, we need to ask for an extension of 1 week.

Minor:

The manuscript is poorly prepared and needs thorough editing. Examples (not comprehensive):
We agree that the manuscript contained some grammatical errors and misspellings words that have been corrected in the present version.

Line 57 "proposed" not propose.
Check

Line 75 "Also" not Although.
Both corrected

Line 80, remove wild card
Removed

Results 2.1 a brief remark should be included that only one type of diet is used.

We have now included this suggested comment.

Table 1: Headings it is not WT and ob/ob but BTBR and BTBR ob/ob.
Corrected

Line 97 line feeds after table are missing.
Data included

Fig. 1. To allow a direct comparison with ref. 21, the liver TG content as mg/g should be provided       
Triglycerides were determined in 5 BTBR and 5 BTBR ob/ob animals at 12 weeks of age whose livers were frozen, obtaining an amount of 13-16.5 mg/g in BTBR and 36-77 mg/g in BTBR ob/ob, similar results obtained for Lan et al.
The problem is that, in order to make an accurate comparative measurement, we think it would be necessary to do the quantifications at the same time, which would require additional time to complete the study. So in case the referee considers this data essential, we need to ask for an extension of 1 week.

Fig. 3. G provide exact explanation how the wild type "1" was determined.
Gene expression were normalized considering the average expression in BTBR WT of each gene as 1, therefore the increase or decrease in gene expression is a n-fold change versus the average expression on BTBR WT, as clarified in the methodology section.

Table 2  it would be useful to determine the glycogen content  in mg/g.
In the manuscript, we mainly focused on the lipid content and unfortunately, the measurement of glycogen after the time elapsed from animal slaughter could bias or obtain erroneous results (because the measurement is based on quantifying free glucose from the cut-off of glycogen stores after liver tissue lysis). To try to overcome this problem, we include a quantitation on PAS section on all animals studied.
Additionally, we do not have a large stock of tissue to perform the experiments on all animals.

Line 170, again line feed after table is missing.
Corrected

Discussion line 250 and subsequent: see above, the discussion of the results of the Lan publication (ref. 21) is completely misleading.
A detailed explanation of why we consider that is not misleading is included in the first reply.

Line 291, the sentence does not make sense.
Removed

-Fig. 7 nice, but absolutely unnecessary and not providing any functional connections. 
This figure was intended as a graphical summary and now it is loaded separately.

-The discussion is rather long and ought to be shortened.
The discussion has been revised and shorten.

Methods, line 370 the conditions under which the animals were kept prior to killing (8 h fast, 4 h fast???) are missing.
All measurements at the time of sacrifice were performed under non-fasting conditions.

4.4 it should be stated, whether the person who did the scoring was blinded for the experimental group.
The reviewer is right in that it has not been reflected in the methods, so a modified paragraph has been added accordingly: “The liver sections were classified according to a semiquantitative histopathological score damage (NAS, NAFLD activity score) performed by trained personnel in a blinded manner”.

Reviewer 2 Report

Abstract

The abstract is well written

Introduction

1-Are the characteristic of MAFLD in human and rats? I think using MAFLD in rodents could be used with precaution

2-Delete 2.results in line 80

Results

1-Please analyze more the results. As example, important metabolic changes parameters is not enough. Indicate some value or percentage.

2-Line 177: what do you mean by “the NAFLD activity score”

3-Why did you not measure arterial blood pressure?

4-Did you evaluate the daily quantity of food intakes?

5-Why did you not perform IPGTT and IPIGT?

Figures

1-Please indicate the staining of F4/80, CD3, MPO, pSTAT3, pNRF2, 4-HNE with arrow on different figures

2-Concerning western blot, why for DGAT2 and FASN, membrane has been separated into two?

Discussion

1-Discuss principally of our results perform in this study. “This model was previously studied by Lan et al. when they compared microarray profile between two experimental models  of diabesity: C57BL/6J (B6) ob/ob (diabetes resistant) and BTBR ob/ob (diabetes susceptible). In this article, they describe that BTBR ob/ob mice at 14 weeks did not develop hepatic steatosis compared to the B6 ob/ob, with downregulation of enzymes and transcription factors related to DNL, to conclude that “Intriguingly, hepatic steatosis is inversely correlated with diabetes susceptibility in obese mice.”, so, synthesize these results

2-As mentioned using the term “mafld” in this animal model should be used with precaution.

Author Response

Dear Reviewer

Responses to your comments and suggestions are attached in the document.
We thank you in advance for your feedback.

Reviewer 3 Report

The authors of the article are thanked for the quality of the work and the choice of the subject relating to the pathophysiological mechanism involved in MAFLD development in Black Tan and brachyuric (BTBR) insulin-resistant mice in combination with leptin deficiency (ob/ob). It is known that  Metabolic associated fatty liver disease (MAFLD) is one of the most common liver disorders worldwide. Opazo-Rios and collaborators discuss the liver morphology and biochemistry on diabetic and obese mice model (BTBR ob/ob) as well as a diabetic non-obese control (BTBR + streptozotocin) and non-diabetic control mice (BTBR wild type) from 4-22 weeks. The manuscript entitled “Meta-inflammation and de novo lipogenesis markers are involved in metabolic associated fatty liver disease progression in BTBR ob/ob model” is a good study, scientifically valid, well executed, and deserve some space in the journal. However, some concerns have been raised. My major concern is that details of methods in Material and Methods section should be provided, as well as other minor concerns:

Page 1, line 45: The abbreviation “T2DM” should be clarified as it did not appear in the text before

Page 12, lines 357-382: The studies carried out are very complicated. While reading the text, I made a diagram that helped me understand the number and names of the groups, as well as  the order of the procedures performed. Please add a figure (diagram, draft or schema) to the section Material and Methods, that shows the model of the experiment. 

Page 13, lines 384-387: Explain the triglycerides (TGs), total cholesterol, high-density lipoprotein cholesterol (HDL), low-density lipoprotein cholesterol (LDL), alaninÄ™ transaminase (ALT), aspartate transaminase (AST), alkaline phosphatase (AP) and albumin preparation technique in detail. Were samples prepared before these tests? If so, how? Include the entire procedure.

Page 13, line 389: Explain the liver lipids extraction technique in detail.

Page 14, lines 438-439: What criteria were taken into account in the histopathological score damage concerning determination of steatosis, lobular inflammation and hepatocytes ballooning in livers? Details of studies should be added.

Figure 2 A, C, E: Immunohistochemistry without a proper negative control assume to be not convincing. Write about negative control and include the section in the plate as well.

Figure 3 A, C, E: Immunohistochemistry without a proper negative control assume to be not convincing. Write about negative control and include the section in the plate as well.

Author Response

(The authors gave the same response as above.)

Reviewer 4 Report

Authors performed an interesting research on MAFLD pathophysiology in animal models. Given the increasing scientific interest for this topic, I think that the paper might add and expand current knowledge in this field.  In my opinion, the study is well designed and conducted.  As minor comments, introduction section might be improved by highlighting the pathogenic role of IR  (e.g. PMID: 32841326) and the cardiometabolic burden of MAFLD since childhood (e .g. PMIDs 34944730;34364544 ) as well.  English language should be finely polished (in particular introduction and discussion sections).

Author Response

Reviewer 4

  • Authors performed an interesting research on MAFLD pathophysiology in animal models. Given the increasing scientific interest for this topic, I think that the paper might add and expand current knowledge in this field.  In my opinion, the study is well designed and conducted.  As minor comments, introduction section might be improved by highlighting the pathogenic role of IR (e.g. PMID: 32841326) and the cardiometabolic burden of MAFLD since childhood(e .g. PMIDs 34944730; 34364544 ) as well.  English language should be finely polished (in particular introduction and discussion sections).

Firstly, we would like to express our sincere appreciation for the reviewer comments and suggestions, which will surely contribute to improve the quality of the manuscript.

A new paragraph has been included pointing to the referee comments:

“Insulin resistance has been acknowledged as a MAFLD predictor in childhood, and both entities associate to an increase cardiometabolic burden since childhood (34944730; 34364544), although the causal relationship among them seems more complex (32841326) as observed in young obese.”

Round 2

Reviewer 1 Report

The authors have made several amendments to the manuscript that adequately deal with the more formal shortcomings of the previous version.  The authors, in their rebuttal, also recognize that the request for the inclusion of additional data to improve the manuscript is not unreasonable but claim that they would need more time to perform the requested studies.

From my point of view, it would be really worth wile to invest this time because this would exploit the full potential of this really interesting study.

Author Response

Dear Reviewer

In the attached document, we reply to their comments.

Thanks for all your feedback.

Reviewer 2 Report

The manuscript has been strongly improved. And the authors' responses are very satisfactory

Author Response

Dear reviewer

Thanks for your feedback for and thereby improve the quality of the manuscript.

Round 3

Reviewer 1 Report

All concerns adequately dealt with. Nice manuscript!